# The Applications of High-Intensity Focused Ultrasound (HIFU) Ablative Therapy in the Treatment of Primary Breast Cancer: A Systematic Review

**DOI:** 10.3390/diagnostics13152595

**Published:** 2023-08-04

**Authors:** Dania Zulkifli, Hanani Abdul Manan, Noorazrul Yahya, Hamzaini Abdul Hamid

**Affiliations:** 1Functional Image Processing Laboratory, Department of Radiology, Universiti Kebangsaan Malaysia, Jalan Yaacob Latif, Bandar Tun Razak, Cheras, Kuala Lumpur 56000, Malaysia; zulkifld@tcd.ie (D.Z.); hamzaini@ppukm.ukm.edu.my (H.A.H.); 2Department of Radiology and Intervency, Hospital Pakar Kanak-Kanak (Children Specialist Hospital), Universiti Kebangsaan Malaysia, Jalan Yaacob Latif, Bandar Tun Razak, Kuala Lumpur 56000, Malaysia; 3Diagnostic Imaging and Radiotherapy Program, Centre for Diagnostic, Therapeutic and Investigative Studies (CODTIS), Faculty of Health Sciences, Universiti Kebangsaan Malaysia, Jalan Raja Muda Abdul Aziz, Kuala Lumpur 50300, Malaysia

**Keywords:** HIFU, breast cancer, necrosis

## Abstract

Background: This study evaluates the role of high-intensity focused ultrasound (HIFU) ablative therapy in treating primary breast cancer. Methods: PubMed and Scopus databases were searched according to the PRISMA guidelines to identify studies from 2002 to November 2022. Eligible studies were selected based on criteria such as experimental study type, the use of HIFU therapy as a treatment for localised breast cancer with objective clinical evaluation, i.e., clinical, radiological, and pathological outcomes. Nine studies were included in this study. Results: Two randomised controlled trials and seven non-randomised clinical trials fulfilled the inclusion criteria. The percentage of patients who achieved complete (100%) coagulation necrosis varied from 17% to 100% across all studies. Eight of the nine studies followed the treat-and-resect protocol in which HIFU-ablated tumours were surgically resected for pathological evaluation. Most breast cancers were single, solitary, and palpable breast tumours. Haematoxylin and eosin stains used for histopathological evaluation showed evidence of coagulation necrosis. Radiological evaluation by MRI showed an absence of contrast enhancement in the HIFU-treated tumour and 1.5 to 2 cm of normal breast tissue, with a thin peripheral rim of enhancement indicative of coagulation necrosis. All studies did not report severe complications, i.e., haemorrhage and infection. Common complications related to HIFU ablation were local mammary oedema, pain, tenderness, and mild to moderate burns. Only one third-degree burn was reported. Generally, the cosmetic outcome was good. The five-year disease-free survival rate was 95%, as reported in two RCTs. Conclusions: HIFU ablation can induce tumour coagulation necrosis in localised breast cancer, with a favourable safety profile and cosmetic outcome. However, there is variable evidence of complete coagulation necrosis in the HIFU-treated tumour. Histopathological evidence of coagulation necrosis has been inconsistent, and there is no reliable radiological modality to assess coagulation necrosis confidently. Further exploration is needed to establish the accurate ablation margin with a reliable radiological modality for treatment and follow-up. HIFU therapy is currently limited to single, palpable breast tumours. More extensive and randomised clinical trials are needed to evaluate HIFU therapy for breast cancer, especially where the tumour is left in situ.

## 1. Introduction

Breast cancer is one of the most typical types of cancer globally. It is the second leading cause of cancer deaths in women [1], with an incidence rate of 10.4% of all cancers [2]. Hence, breast cancer has become one of the most well-researched and studied malignancies in oncology. The incidence rate of female breast cancer has been slowly increasing by 0.5% per year since the mid-2000s, likely due to a decrease in the fertility rate and an increase in excess body weight [1], and also because of the rise in population awareness and the success of breast cancer screening programs. Breast cancer has been increasingly diagnosed at an earlier stage. The current evidence-based management of breast cancer involves surgery, radiotherapy, chemotherapy, and endocrine therapy. In recent decades, the management of early breast cancer has evolved from radical mastectomy to breast conservation surgery. This change towards breast conservation surgery has not affected the long-term survival rates of breast cancer patients [3].

The advancement in medical technology has allowed for the development of new and improved treatments. The field of medicine has shifted its focus towards maximising the utilisation of technology to produce medical treatments that are efficient and minimally invasive and provide superior or equivalent clinical outcomes. Interventional oncology is an exciting and promising branch of interventional radiology, where minimally invasive therapies can be integrated into an individualised multidisciplinary oncological care plan, alongside medical, surgical, and radiation oncology. Examples of interventional oncology include cancer therapies, i.e., chemoembolisation and ablation [4]. The treatment guidelines have successfully included ablative therapy and chemoembolisation for treating hepatocellular carcinoma [5].

As an alternative to surgical resection, thermal ablation destroys viable cells of the targeted tissue volume by extreme hyperthermia or hypothermia [6]. Non-surgical, minimally invasive ablative techniques, i.e., microwave ablation, cryoablation, laser ablation, radiofrequency ablation, and high-intensity focused ultrasound (HIFU), have been studied for breast cancer treatment. These ablative therapies can be offered as options for a selected type of breast cancer, as local treatment for small primary tumours or residual tumours after systemic therapy, or as a salvage method to treat local recurrence after breast conservation therapy [3].

High-intensity focused ultrasound (HIFU), one of the more active research techniques, has proven to be an effective non-invasive, non-ionising ablative therapy for soft tissue [7]. HIFU ablation aims to deliver sufficient energy to increase tissue temperature to a cytotoxic level, causing coagulation necrosis [7]. Both ultrasonography (US) and magnetic resonance imaging (MRI) can be used as image-guidance modalities for HIFU therapy. The clinical applications of HIFU therapy have been researched for both benign and malignant conditions, i.e., uterine fibroids, fibroadenoma, prostate cancer, breast cancer, and liver cancer. HIFU therapy has been proposed as an alternative for many surgical procedures. This also includes breast conservation surgery for localised breast cancer.

Therefore, this systematic review aims to explore and evaluate the role of HIFU ablative therapy in primary breast cancer treatment. The safety, efficacy, and feasibility of HIFU therapy and its clinical and technical challenges will be addressed in this study.

## 2. Methodology

### 2.1. Eligibility Criteria

The systematic review was conducted based on the Preferred Reporting Items for Systematic Reviews and Meta-Analyses (PRISMA) 2020 guidelines [8] and followed previous studies [9,10,11,12,13,14,15,16]. The PICOS criteria were used to determine the inclusion and exclusion criteria for the article selection based on population, intervention, comparison, outcome, and study design, as in Table 1. The inclusion criteria were women aged 18 and above with primary breast cancer who received HIFU therapy in addition to the conventional multimodal breast cancer treatment regimen. Clinical trials and case-control, cohort, and cross-sectional studies were included. Exclusion criteria were men, children, metastatic breast cancer, other types of cancer, other types of thermal ablation therapy, non-clinical outcomes, and studies from more than 20 years ago. The articles that fulfilled all five criteria were deemed relevant for the present study.

### 2.2. Information Sources and Search Strategy

The electronic search databases used were PubMed (National Centre of Biotechnology Information) and Scopus to identify the most recent articles up to 20 years prior. The study search started on 27 October 2022 and was completed on 10 November 2022. The keywords used were *((((BREAST CANCER) OR (BREAST CARCINOMA)) OR (BREAST NEOPLASM)) AND (((HIFU) OR (HIGH INTENSITY FOCUSED ULTRASOUND)) OR (ABLATION))) AND ((((BREAST CONSERVING SURGERY) OR (LUMPECTOMY)) OR (WIDE LOCAL EXCISION)) OR (SURGERY))*. The non-English and non-human articles were excluded. A total of 1535 articles were generated from the databases; 465 articles were obtained from PubMed, and 1070 were obtained from Scopus. These articles were transferred onto Google Sheets to aid with the screening process (Figure 1).

### 2.3. Selection Process

The 1535 articles were independently screened based on the title and abstract, which excluded 1518 articles. Seventeen articles were retrieved for full-text screening. A total of eight articles were further excluded. The final nine articles were reviewed and confirmed by Hanani Abdul Manan. These articles were evaluated, and relevant essential information was presented in a table to include in the study. The search strategy based on PRISMA and selected articles is tabulated in Figure 1.

### 2.4. Data Extraction

Data from the nine studies were extracted and tabulated based on the domains—author(s), year, country, study design, demographic information, HIFU image guidance modality, type of breast cancer, tumour size, time of surgical resection after HIFU, clinical evaluation or HIFU complication, radiological evaluation, pathological evaluation, and follow-up outcomes. The data are tabulated in Table 2 and Table 3.

## 3. Results

### 3.1. Study Selection

The database searches yielded 465 articles and 1070 articles from PubMed and Scopus, respectively. The 1535 articles were independently screened based on the title and abstract, which excluded 1518 articles. Seventeen articles were retrieved for full-text screening. A total of eight articles were further excluded. Four out of the eight articles were excluded after full-text screening due to unmet PICOS criteria. These four articles did not include the clinical outcome of the study and did not match our inclusion criteria for the present study. Two of the eight articles were systematic reviews and meta-analyses, and two out of the eight articles were duplicates. Therefore, nine articles were selected in the present study, which fulfilled our PICOS. The details of the selected studies are presented in Figure 1.

### 3.2. Study Characteristics

The nine selected studies’ publication dates were between 2003 to 2016, and most of the studies are from China (five out of the nine studies), by Wu et al. [21,22,24,25] and Guan et al. [17] from China; others include one study from Israel [18], one study from Japan [19], one study from Canada [20], and one study from the Netherlands [23]. Four of the nine studies were from the same author from Chongqing University of Medical Sciences, Chongqing, China, by Wu et al. [21,22,24,25]. Two of the three studies by Wu et al. [22,24] were published in 2006 and 2007 and used the same sample size of 23 patients.

The study designs include two randomised control trials (RCTs) [17,25] and seven prospective studies [18,19,20,21,22,23,24]. The RCTs compared groups of patients receiving both HIFU therapy and mastectomy to mastectomy only. The number of patients recruited in the RCTs was 50 patients and 48 patients, respectively. The prospective studies involved 130 patients who received HIFU therapy before and alongside the conventional multimodal treatment for breast cancer, i.e., surgery, radiotherapy, chemotherapy, and endocrine therapy. There may be a possible overlap of patients from three of the studies by the same author Wu et al. [21,22,24], from Chongqing, China, resulting in the overestimation of the total patients involved. The sample size range is between 10 to 50 (median of 23), while the mean age range is between 45 and 60. The population of the present study was adult women with primary breast cancer, who had histologically proven solitary tumours. The tumour size varied from 0.5 to 6.0 cm. All of the studies’ stages of breast cancer varied, and most patients had early-stage breast cancer, stages 1 and 2.

The outcome of the HIFU therapy was assessed clinically, radiologically, and pathologically (please refer to Table 3). Skin changes, including burns and pain, were clinically evaluated post-HIFU treatment, while tissue changes, both necrosis of targeted tissue and effects on surrounding tissue, were radiologically evaluated. Most studies used ultrasound or MRI to assess breast tissue post-HIFU, but other radiological modalities that were included in the studies were colour Doppler ultrasound and SPECT. All histopathological assessments of treated tissue used haematoxylin and eosin (HE) staining for tumour necrosis. Others included UEAI staining, NADH-diaphorase staining, Victoria blue and Ponceau’s histochemical staining, TTC staining, TUNEL method, and immunohistochemical staining.

The survival outcomes were measured in two of the studies by Guan et al. [17] and Wu et al. 2005 [21] to determine the 5-year disease-free survival rate, rate of local recurrence, and distance metastasis rate. The study by Guan et al. showed a 95% 5-year disease-free survival rate, a 9% rate of local recurrence, and no distance metastasis. Wu et al. 2005 [21] reported a 95% 5-year disease-free survival rate and an 89% 5-year recurrence-free survival rate.

### 3.3. Type of HIFU Device for Treatment of Breast Cancer

Most studies (five out of nine) used ultrasound-guided HIFU devices, while the remaining studies used MR-guided HIFU devices. The five studies conducted in China used the JC-HIFU therapeutic system (Chongqing HAIFU Technology Company, Chongqing, China). The studies from Japan, Israel, and Canada used the MR-guided focused ultrasound surgery (MRgFUS) ExAblate 2000 system (InSightec, Tirat Carmel, Israel). The study from the Netherlands used a MR-HIFU breast platform (Sonalleve-based prototype, Philips Healthcare, Vantaa, Finland). Gianfelice et al. [20] used two US ablation systems: Mark 1 focused-US system and Mark 2 focused-US system. The advantages of the Mark 2 system (Insigntec-TxSonic) are a multiple-element phased array transducer capability to treat deeper lesions inside the body and up to 20 cm versus Mark 1, which is up to 10 cm, as well as angular motion in the Y mechanical positioning, capability to generate thermal maps, reporting capability, and automated imager interface.

### 3.4. Suitability Criteria for HIFU Therapy

#### 3.4.1. General Criteria

Patients with significant medical conditions, e.g., cardiovascular disease, COPD, congestive heart failure, coagulation disorders, and poorly controlled diabetes mellitus, were not included in most of the studies. Lactating or pregnant patients and patients on anticoagulation were also deemed unsuitable [18,19]. Other exclusion criteria were metallic implants, other incompatibilities with MRI, inability to lie still for up to 150 min [19], and breast implants [19,25].

#### 3.4.2. Tumour Criteria

Patients with pathology-proven invasive breast cancer were selected for HIFU therapy. Suitability criteria for HIFU therapy were the following: single palpable tumour less than 5 cm in diameter, the lesion boundaries visualised with colour Doppler ultrasound imaging, circumscribed at least more than 1 cm from the skin or ribcage, and more than 2 cm from nipple [17]; tumour easily visible on non-contrast MR, more than 1 cm from the skin and chest wall, and lesion size not greater than 3 cm with no diffuse microcalcifications [18]; lesion size less than 3 cm, more than 1 cm from the skin, nipple, or ribcage, more than one focal breast lesion per quadrant, an invasive intraductal component, and previous radiation or local thermal therapy [19]; lesion size less than 3.5 cm and a minimal distance of 1 cm between the lesion and the skin and the ribcage [20]; number of lesions 2 or less, size 5 cm or less with boundaries visible on US imaging, distance from skin or ribcage at least 0.5 cm and distance from nipple at least 2 cm [21]; single palpable tumour, lesion size of 6 cm or less with boundaries visualised on US imaging, distance from skin or ribcage at least 0.5 cm and distance from nipple more than 2 cm [22,24]; maximum tumour diameter of 1 cm or more, tumour location within the reach of HIFU transducers with the patient in prone position, 1 cm or more distance from skin and pectoral muscle to the centre of the target [23]; single palpable tumour not greater than 6 cm in diameter with lesion boundaries visualised with colour doppler US imaging, at least more than 0.5 cm from skin or ribcage, and more than 2 cm from nipple [25].

Most of the studies included patients with single palpable tumours except the study by Gianfelice et al. [20], which included patients with calcifications. Additionally, Wu et al. [21] inclusion criteria were patients refusing modified radical mastectomy or those unsuitable for surgical resection.

### 3.5. Clinical and Radiological Evaluation of Breast after HIFU Therapy

#### 3.5.1. HIFU Therapy Complications

Non-severe HIFU complications noted were local mammary oedema, warmth, a sensation of heaviness in the treated breast, and mild local pain, discomfort, and tenderness. Some of the patients required analgesia to manage the pain.

Skin burns were observed in four of the studies. These were third-degree burns [19], second-degree skin burns [18,20], and minimal skin burns [25]. In regard to cosmetic outcomes, Wu et al. [21] used a five-point scale for cosmetic evaluation, which showed 96% good to excellent and 6% acceptable.

Other complications were pyrexia over 38 degrees Celsius for 48 h [17]; claustrophobia; allergic reaction to the plastic material of the patient’s interface; shoulder pain [19]; self-resolving benign small white lump; nausea and vomiting [23]; and palpable lump larger than lesion size [24].

Finally, no studies observed severe complications such as infection or haemorrhage.

#### 3.5.2. US Imaging and MRI Evaluation

In the study by Furusawa et al. [19], MR images were taken immediately following HIFU therapy. The T1-weighted contrast-enhanced subtraction image post-HIFU therapy showed that the tumour was non-enhancing, with hyperintense areas in the edges of the treated region (hyperaemia).

Wu et al. [21] conducted an extensive radiological evaluation follow-up post-HIFU therapy, to evaluate the therapeutic efficacy of HIFU therapy and to detect evidence of a residual tumour in the treated region or growth of a new tumour in the diseased breast. All patients had pre- and post-HIFU therapy colour Doppler US imaging. Additionally, six patients and five patients had SPECT and MRI, respectively.

In the Wu et al. [21] study, B-mode US showed a heterogenous increase in grey-scale within the treated lesion after HIFU therapy in 15 out of 22 patients, but no grey-scale change was reported in the remaining 7 patients. Colour Doppler US imaging showed no blood flow within the treated lesion in 19 out of 22 patients. The US imaging was used to assess the reduction in tumour size during the follow-up period at 3, 6, 12, 24, 36, 48, and 60 months. The tumour disappeared in 8 patients, and size reduction was seen in 14 patients [21].

Post-HIFU therapy MRI evaluation in five patients showed either increased or decreased signal on both T1- and T2-weighted unenhanced MRI sequences. Gadolinium-enhanced MRI showed an absence of contrast enhancement in the treated lesion and a thin peripheral rim of enhancement, indicative of coagulation necrosis. Six patients who had Tc-99m sestamibi SPECT showed that the radioisotope uptake disappeared in the treated lesion, indicating an absence of viable tumour cells after HIFU therapy. These findings were reported by Wu et al. [21].

Merckel et al. [23] reported that there were no visual differences observed between contrast-enhanced MRI before and immediately after HIFU therapy. In Wu et al. [24] and 2003, three patients underwent unenhanced and gadolinium-enhanced MRI evaluations. The unenhanced MRI showed slight signal changes on both T1- and T2-weighted images. The enhanced MRI showed an absence of contrast enhancement in the treated lesion, including breast cancer and the surrounding 1.5–2 cm normal breast tissue, indicating the extent of coagulation necrosis after HIFU therapy.

### 3.6. Histopathological Evaluation of Breast after HIFU Therapy

Gross (macroscopic) evaluation of the breast tissue post-HIFU therapy and surgical resection showed thorough coagulation necrosis of the targeted tissue, which included the tumour and average margin of 1.92 ± 0.35 cm (between 1.8 and 2.3 cm) of normal breast tissue surrounding the tumour [17] and the presence of central necrosed tissue (yellow-white area) surrounded by haemorrhagic tissue (red area) [20].

For microscopic evaluation, haematoxylin and eosin (HE) staining were used in all studies to evaluate tumour necrosis after HIFU therapy. HIFU therapy showed immediately caused cell damage, uniform coagulative necrosis of the entire target tumour, and irreversible cell necrosis 24 h after HIFU therapy, which is evident in the post-procedural biopsy in all patients (100%) [17]. Similarly, coagulation necrosis was observed macroscopically in all patients and microscopically in 21 out of 23 patients (91.3%) [22]; in addition, tumour necrosis was absorbed in 6 out of 10 patients (60%) by tissue coagulation and leakage of erythrocytes [23].

Furthermore, Wu et al. [24] reported no TTC uptake and no TUNEL-positive cells observed in any of the HIFU-ablated tumour cells. This result indicated complete destruction of targeted breast cells by coagulation necrosis with apoptotic cells observed [24]. Additional immunohistochemical staining used by Wu et al. [25] showed complete coagulation necrosis in targeted cells with an absence of malignant behaviours, i.e., proliferation, invasion, and necrosis. There was no expression of molecular indicators representing malignant behaviours, e.g., PCNA, CD44v6, and MMP-9 in HIFU-treated tumours compared to the control group [25].

For the remaining studies, the results of the tumour necrosis were quantified. Furasawa et al. reported that 95% achieved 100% necrosis, and 10% had less than 95% necrosis. The mean necrosis of an ablated tumour was 96.9%. Gianfelice et al. [20] reported that 17% achieved 100% necrosis. There was 43.3% mean necrosis in Mark 1 group and 88.3% mean necrosis in Mark 2 group. Wu et al. 2005 [21] evaluated the HIFU-ablated tumour tissue at 2 weeks, 3 months, 6 months, and 12 months, which showed no viable tumour cells or coagulative necrosis after 2 weeks, partial fibrosis after 3 months, and complete necrosis after 6 months and 12 months. In the study by Zippel et al., only two patients (25%) had a complete pathological response with 100% tumour necrosis achieved, while eight patients had varying amounts of residual tumour.

### 3.7. Follow-Up Outcomes of HIFU Therapy

In two of the nine studies, Guan et al. and Wu et al. [21], conducted a yearly follow-up for 5 years. Guan et al. reported a 95% 5-year disease-free survival rate, a 9% rate of local recurrence (*n* = 2), 5% death (*n* = 1), and no distant metastatic lesions. The study by Wu et al. 2005 [21] had a median follow-up of 54.8 months (36 to 72 months). The disease-free and recurrence-free survival rates at 1, 2, 3, 4, and 5 years were 100% and 100%, 100% and 89%, 100% and 89%, 95% and 89%, 95% and 89%, respectively. Overall, there were 95% 5-year disease-free, 89% 5-year recurrence-free survival, and 5% death rates (*n* = 1). Two out of twenty-two patients, both stage IIb, developed local recurrence at 18 and 22 months post-HIFU therapy. Both underwent modified radical mastectomy and chemotherapy. One of the two patients developed lung metastasis 37 months post-HIFU therapy (19 months after mastectomy) and died of extensive metastases at 44 months post-HIFU therapy.

## 4. Discussion

This study aimed to evaluate the role of high-intensity focused ultrasound (HIFU) therapy in breast cancer treatment. The results from the nine studies have shown that US-guided HIFU or MRI-guided HIFU caused coagulation necrosis of breast cancer tumours. Most patients achieved necrosis of 100% with minimal side effects and good cosmesis.

### 4.1. Safety and Efficacy of HIFU Therapy for Clinical Usage for Breast Cancer

The studies have shown that HIFU ablation therapy induces coagulation necrosis in primary breast cancer, which is evident in radiology and histopathology. HIFU-induced coagulation necrosis of the targeted tumour was replaced entirely by fibroblastic scar tissue, followed by gradual resorption of the fibroblastic tissue [21]. The histological evaluation also showed the absence of malignant behaviours, i.e., proliferation, invasion, and metastasis [25]. Post-HIFU therapy US evaluation showed a reduction in tumour size or complete disappearance of the tumour, and a heterogenous increase in grey-scale within the treated lesion. Also, the absence of blood flow within the treated lesion on colour Doppler US and the absence of contrast enhancement in the treated lesion on enhanced MRI indicates coagulation necrosis.

Clinical evaluation by Guan et al. [17] reported changes in the grey-scale of the targeted tissue on US post-HIFU ablation, where the hyperechoic region corresponded well to the extent of coagulation necrosis. There was a positive relationship between the hyperechoic extent measured in US and the extent of necrosis measured on gross examination, both in vivo and in vitro.

However, not all HIFU therapy patients achieved a complete pathological response. Also, the percentages of necrosis for the ablated breast tissue varied. This may suggest that residual cancer tissues are still possible although HIFU ablation effectively kills breast cancer tissues. The evidence of the efficacy of HIFU ablation for breast cancer treatment to date has been variable [17]. Overall, the disease-free survival rate is 95%, in the patient group who received HIFU therapy before surgery and other conventional breast cancer therapies. This result was reported in both of the RCTs included in this study.

Many factors affect the efficacy of HIFU ablation. Technical factors, i.e., HIFU lesion area size, focal region sound intensity, irradiation time, and amount of irradiation, and tumour factors, i.e., smaller blood vessels with a diameter of 2 mm or less, are all positively correlated with more obvious damage by HIFU [17]. The number of locations targeted by sonication was found to be equal to the number of areas with tumour necrosis on histopathology [23]. Furthermore, there was a good correlation between the power of sonication and the size of tumour necrosis on histopathology [23]. Gianfelice et al. [20] suggests treating the cancerous tumour plus a 10 mm layer of boundary tissue to ensure total thermal coagulation necrosis of the cancer zone and the clusters at the periphery.

In terms of safety, most of the studies reported non-severe side effects from the HIFU ablation. Immediate localised side effects such as oedema, warmth, and pain were common and expected from the therapy. The oedema was likely to subside within 3–10 days [19,20], and the pain was manageable with simple analgesia. Five patients from the studies experienced skin burns, which were mostly mild to moderate. Furusawa et al. [19] reported that a third-degree skin burn was likely caused by high-energy sonication close to the skin. Burns complications may be further minimised with the development of a cooling system [18]. Long-term complications were not reported in the studies. The imaging used to guide HIFU therapy, ultrasound, and MRI does not pose significant patient risks. Overall, HIFU ablation is a non-invasive targeted therapy that is generally safe.

Other side effects reported in the studies are related to the overall procedure and not directly from the HIFU ablation. These side effects include nausea and vomiting, likely anaesthetic-induced [23]; pyrexia over 38 degrees Celsius for 48 h [17]; claustrophobia; allergic reaction to the plastic material of the patient’s interface; and shoulder pain, which was positional in nature [19].

Guan et al.’s findings concluded that HIFU ablation is a non-invasive treatment that can kill tumour cells and completely stop the proliferation of blood vessels while preserving the surrounding normal tissues. Unlike anti-angiogenic agents, e.g., Bevacizumab, where cancer cells can escape and develop resistance to anti-angiogenic agents, HIFU ablation causes complete coagulation necrosis of breast cancer tissue which may destroy the interdependence of tumour cell proliferation and blood vessel formation, disrupting the proliferation cycle.

### 4.2. Selection of Breast Cancer Patients Suitable for HIFU Therapy

Most of the studies had strict inclusion criteria for patient selection. Distance between the tumour and the skin and the nipple are essential factors that need to be established before HIFU ablation to effectively target the tumour and minimise the risk of skin burns and other related side effects. The minimal distance between the tumour and the skin should be at least 0.5 cm to 1 cm, and the minimal distance between the tumour and the nipple should be at least 2 cm. This range was consistent across all the studies. Tumours close to the skin are more likely to cause skin burns and leave residual cancer cells if the overlying skin is undamaged [24,25].

Most of the studies only included patients with a solitary breast tumour, which was palpable, and tumour margins visualised on imaging. In general, the tumour size was less than 5 cm in diameter. Larger tumours or large T2 tumours were not considered appropriate for ablative therapies, e.g., HIFU therapy [23]. Small tumours with less than 1 cm^3^ in volume can be eradicated entirely by a single irradiation pulse, but larger tumours will require multiple HIFU irradiation [17]. Larger tumours will necessarily require an additional amount of treatment time [18]. The number of sonication points depends on the tumour’s size and the focal point volume [20]. The tumour size also influences the cosmetic outcome [21].

The role of HIFU therapy for multi-focal or non-palpable breast tumours and axillary lymph nodes is unknown due to limited evidence from the current experiments. Overall, the current protocol may limit the application of HIFU therapy to small, solitary, and palpable peripheral breast tumours. Wu et al. [25] suggested excluding breast tumours with undefined margins, scattered multiple foci, and located at close proximity to the nipple for potential HIFU therapy.

### 4.3. Advantages and Disadvantages of HIFU Therapy for Breast Cancer

The benefits of HIFU therapy for breast cancer include lack of infection, no bleeding, preserving the structure, function, and integrity of breast tissue, no scarring, minimal change to breast shape, and good cosmetic outcome. HIFU therapy is non-incisional, as it does not require probe insertion. HIFU therapy, with its non-invasiveness, increasing effectiveness, and good safety profile, has a potential application for breast cancer treatment.

Interstitial laser therapy, cryotherapy, and radiofrequency ablation require probe insertion, and the probe position is fixed to the focal point. In interstitial laser therapy, most of the energy is delivered at the tip of the probe; relying on thermal diffusion to spread the energy to a larger volume and requires longer ultrasound beam exposure times. In comparison with other thermal ablation techniques, HIFU therapy is highly flexible since the focal point is changeable—the size and shape of the treatment zone can be controlled to match the target volume [20] correctly. Focused US sonication is achieved in a few seconds, resulting in local temperature elevations independent of the perfusion rate and reproducible from location to location [20]. Also, HIFU therapy has the advantage of being repeatable.

The clinical effectiveness of HIFU therapy will be dependent on the imaging technical monitoring and guidance either by using ultrasonography (US) or magnetic resonance imaging (MRI) [17]. Ultrasonography is accessible, cost-effective, flexible, and mechanically compatible, whereas MRI offers temperature measurement and high-resolution images [17]. In comparison with other potential ablative therapies, i.e., cryotherapy, laser and radiofrequency ablation, HIFU allows for real-time monitoring, closed-loop feedback, and non-invasiveness [18,19].

Without interrupting skin integrity, MRI-guided HIFU can precisely deliver energy with up to 1 mm accuracy to a given point in soft tissue, and the operator is fully in control of the induced thermal effect based on the real-time feedback detailing temperature changes [19]. The detection of temperature changes feature in MRI-guided HIFU is important, as it enables rapid evaluation of dosimetric parameters. Also, the high sensitivity of MRI is beneficial in accurately targeting breast cancer [20].

Most patients in the studies underwent surgical resection post-HIFU therapy, and the resected tissues were histologically assessed to evaluate tumour necrosis. The findings were variable—not all patients achieved complete necrosis with HIFU therapy. Retrospective analysis by Furusawa et al. [19] claimed that the recommended treatment margins were not respected in patients who did not achieve 100% tumour necrosis. Conversely, Gianfelice et al. [20] reported that the residual tumours that were found in patients who did not achieve 100% tumour necrosis were mainly at the periphery of the treatment field; likely because only 5 mm layer of boundary tissues alongside the tumour were ablated instead of 10 mm layer of boundary tissue.

In Merckel et al., no tumour necrosis was observed post-HIFU therapy in three of the ten patients. This was due to suboptimal treatment and technical factors related to HIFU ablation. For example, only one instance of therapeutic sonication was performed which was stopped soon after its initiation; sonication was targeted at the adipose tissue which was anterior to the tumour because the tumour was outside the range of the transducers, and there was an incorrect focal point target of the tumour due to incorrect misalignment correction after the test sonication.

To accomplish complete (100%) tumour necrosis, HIFU ablation should target the cancerous tumour plus 10 mm layer of boundary tissue by using more focal points, or by increasing the level of energy delivered to each point while keeping the interpulse delays long enough to avoid skin damage [20]. The precision of the focused sound waves needs to be perfected to be able to destroy the maximal amount of tumour tissue within the lesion while having a margin of surrounding healthy tissue (Zippel et al.). Once the precision of HIFU is refined to greater accuracy, HIFU therapy may be beneficial for patient cohorts who are unable or unwilling to undergo surgery, for advanced or recurrent tumours after previous surgery or radiation, or for diffuse and not just discrete tumours [18].

The long treatment time is one of the disadvantages of HIFU therapy. The treatment time for one HIFU therapy session is approximately 120 min [18] or 145 min [23], and ranges between 35 to 133 min [20], during which time the patient will need to be stationary in the MRI machine. The actual sonication time is 1.7 min [23]. The time distribution of MRI-guided HIFU by Merckel et al. [23] is shown in Table 4. The factors affecting the treatment time are the number of sonication points, the sonication time, and the cooling time [20]. Larger tumours require longer treatment times. This may be psychologically challenging for patients who are claustrophobic and anxious. Furthermore, most of the studies had the patients sedated or undergoing general anaesthesia for the HIFU therapy sessions, which may impose an additional cost, unwanted risks, and may appear less desirable to patients. The use of local anaesthesia or intravenous sedation may be more feasible for HIFU ablation; therefore, it should be the choice of anaesthetic method for this procedure in the future [22].

Wu et al. [21] found that all patients had an increase in lesion size immediately following HIFU ablation, likely due to ablation of a volume that was larger than the tumour. Especially in patients with larger tumour size, this increased lesion size after HIFU ablation was more significant, and the resorption of the larger tumour was slower than smaller tumours. Patients who experienced this large thermal-induced lesion post-HIFU ablation were unsatisfied with the outcome, and some resorted for mastectomy. Also, this larger lesion may interfere with the follow-up radiological evaluation to identify early local recurrence [24]. Hence, HIFU therapy may be more preferable and suitable for patients with smaller tumours.

In terms of HIFU therapy’s cost-effectiveness and accessibility, the studies included did not discuss this aspect in detail. For low- and intermediate-risk prostate cancer, active surveillance is more cost-effective than HIFU, but with high uncertainty [26]. In contrast, the use of HIFU as an alternative treatment for painful bone metastases might be cost-effective despite the considerable uncertainty [27]. Further exploration is needed regarding the cost-effectiveness of HIFU therapy for the treatment of breast cancer. Additional training is also needed for radiologists, operators, and other medical professionals to navigate the HIFU device and to establish a suitable treatment protocol.

Overall, HIFU therapy provides advantages due to its non-invasiveness, high efficacy with the correct protocol and monitoring, good safety profile, and good cosmetic outcome. The limitations of HIFU therapy at present are its long treatment duration, narrow and specific breast tumour selection criteria, and potentially high cost. Further development in HIFU technology may provide greater precision with lower treatment time and include a wide range of breast tumours. Treatment protocols should be standardised for HIFU therapy to become more attractive and clinically applicable.

### 4.4. HIFU Therapy in Relation to the Current Multimodal Breast Cancer Treatment

At present, breast cancer management follows national standardised guidelines. Histologically proven malignant breast tumours typically undergo resection, either by wide local excision (WLE)/breast-conserving surgery or mastectomy, followed by radiotherapy, chemotherapy, and endocrine therapy based on the tumour characteristics and classification of breast cancer based on its grade and stage. The current studies on HIFU therapy for breast cancer discussed whether HIFU ablation can either replace breast-conserving surgery, provide an alternative for breast-conserving surgery, or become a conventional treatment modality for breast cancer management.

For HIFU therapy to be comparable to surgical resection, HIFU ablation should achieve complete (100%) tumour necrosis. The current studies showed variable results of HIFU ablation on breast cancer—complete tumour necrosis achieved in 20 to 100% of patients [28]. Zippel et al. suggested HIFU therapy as an adjunctive treatment for patients at high surgical risk or unwilling to undergo surgery, and for patients with advanced and recurrent cancer after previous surgery or radiotherapy.

Sentinel lymph node (SLN) biopsy is an important investigation conducted as part of the breast cancer work-up to determine the stage and treatment pathway. It is unclear if HIFU ablation can alter the lymphatic anatomy to preclude SLN biopsy [18]. Sentinel lymph node biopsy can be performed after ablation therapy [21,29] although it is also possible for it to be performed prior to HIFU ablation [18]. In the study by Zippel et al. [18], two SLN biopsies were performed successfully in patients after HIFU ablation. The characteristic of positive axillary lymph nodes is similar to multi-focal breast cancer, and no published reports are utilising HIFU ablation for axillary lymph nodes [17]. Therefore, axillary lymph node dissection is still necessary after HIFU therapy in the case of positive SLN biopsy, counteracting HIFU therapy’s non-invasiveness [24].

In the studies, several enrolled patients were excluded because they preferred to receive conventional mastectomy, and several patients who had received HIFU therapy opted for further modified radical mastectomy due to fear of residual cancer [21]. Wu et al. [21] reported that 2 of the 22 patients with initial tumour sizes of 3.8 cm and 4.1 cm, respectively, had an increase in tumour size of 6.5 cm post-HIFU therapy. This increase in size impacted the patients psychologically, who subsequently decided to have a mastectomy. Wu et al. [21] suggested that neo-adjuvant chemotherapy be given prior to HIFU therapy for patients with large tumour size in order to decrease size prior to HIFU therapy.

Radiotherapy is given following breast-conserving surgical resection, significantly reducing the recurrence risk. Similarly, the use of adjunct radiotherapy is recommended after HIFU therapy to minimise the risk of local recurrence by targeting the tumour bed [18]. The accuracy and precision of MR-guided HIFU may allow for tumour bed irradiation only while preserving the surrounding healthy breast tissue [18]. Wu et al. [21] suggested that delayed radiotherapy be given once the tumour has regressed or disappeared post-HIFU therapy, as there is currently little evidence of whether radiation of a large volume of necrosis causes severe fibrosis.

In terms of radiological evaluation post-HIFU therapy, Wu et al. [21] suggested using MRI as anatomical imaging and SPECT as functional imaging. The combination of this imaging would be the ideal follow-up modality to determine the therapeutic effects of HIFU on tumour vascularity, cellular viability of breast cancer, and local recurrence [21]. However, MRI and SPECT are unlikely to be cost-effective choices. Ultrasound is cheaper and more accessible; however, it is less sensitive when it comes to assessing coagulation necrosis compared to MRI.

HIFU therapy can be an effective breast-conserving treatment option for breast cancer patients with high surgical risk [7]. HIFU therapy is currently used in the treatment of both benign and malignant tumours [7,30]. The Food and Drug Administration (FDA) has approved HIFU therapy for the treatment of both prostate cancer and bone metastases. The 10-year specific survival rate of prostate cancer patients after HIFU therapy ranges from 92% to 99% [31]. The usage of HIFU therapy for uterine fibroids was approved by the FDA in 2004, with potential for fertility preservation and reduced recovery time [7,32]. HIFU therapy is also being studied for benign breast tumours, i.e., fibroadenoma, showing positive outcomes. The HIFU-F trial concluded that HIFU is feasible for reducing both treatment time and treatment time for breast fibroadenoma [33].

### 4.5. Limitations of the Present Study and How to Improve Future Studies

The studies included in this present study have a small sample size, ranging from 10 to 50 patients, due to specific inclusion and exclusion criteria for patient selection. This represents only a small group of breast cancer cases. Also, presumably the same patient sample was used in the studies by Wu et al. [21,22,24]. There are only two randomised control trials (RCTs) for the past 20 years, and the most recent study was published in 2016. A recent RCT conducted by the University of Oxford in 2022, The HIFUB Study [34], aims to assess the efficacy and safety of HIFU therapy in invasive breast cancer. The research paper is yet to be published. Overall, there are a limited amount of large-scale clinical studies on HIFU therapy for breast cancer.

Most of the studies evaluated the therapeutic effects of HIFU therapy on breast cancer based on histopathological data from conventional surgical resection following therapy. This is called the treat-and-resect protocol, i.e., HIFU ablation followed by surgery [17]). The studies evaluated the efficacy and safety of HIFU therapy, adjunctive to other conventional breast cancer therapies. There is no RCT to date independently comparing HIFU therapy to breast-conserving surgery or studying whether HIFU therapy can replace breast-conserving surgery. The role of HIFU therapy in breast cancer management needs further research to evaluate the clinical applicability of HIFU therapy in relation to the conventional multimodal approach.

Compared to the other studies, the study by Merckel et al. used partial ablation design which is more suited to investigate the safety of HIFU ablation. Future studies on the therapeutic effects of HIFU ablation need to have a standardised protocol that ensures complete tumour necrosis is achieved from each HIFU ablation session. The patient selection, ablation margin, treatment plan, image technical monitoring, and guidance must be systematised. Tumour necrosis is currently evaluated using histopathology assessment of the resected ablated tumour. However, if HIFU ablation were to replace conventional surgery, the evaluation of tumour necrosis will likely need to rely on non-invasive radiological data. Few of the studies have included radiological evaluation of the HIFU-treated tumours where the results have shown variable imaging outcomes post-HIFU therapy. Negative margins visualised on imaging do not always represent pathologically negative margins.

The reliability of radiological evaluation to determine tumour necrosis needs to be further assessed. Ultrasonography and MRI both have their advantages and disadvantages for the choice of image modality. A study by Marincola et al. [35] evaluated of treatment efficacy of both US-guided HIFU and MRI-guided HIFU. The study mentioned that ultrasound does not provide quantitative evaluation of thermal accumulation within the ablated tissue or the adjacent structures, but qualitative evaluation is possible with the presence of a hyperechoic region in B-mode ultrasound or contrast-enhanced ultrasound with micro-bubbles or nano-bubbles. This image modality can assess treatment efficacy by providing pre-treatment imaging, enhancing cavitation, and evaluation of thermal ablation during and post-treatment. Contrast-enhanced MRI seemed to be the superior radiological evaluation due its high sensitivity of 89% to 100% in detecting breast nodules. Contrast-enhanced MRI is able to provide temperature monitoring and the use of intravenous contrast agent. Hectors et al. [36] discussed the multiple MRI methods for evaluating HIFU-ablated tissues. The study concluded that several MRI contrast parameters showed sensitivity to HIFU-ablated tissue changes and suggested multiparametric MRI analysis combining the data of several MRI parameters to be used and studied for HIFU treatment evaluation. Multiparametric MRI analysis may provide more complete evaluation of HIFU treatment because different MRI parameters are sensitive to different treatment types.

Image fusion may be the next important modality for real-time and effective guidance in HIFU-treated breast cancers [17]. Image fusion is the process of integrating multimodal images from one or more imaging modalities to enhance the quality of an image [37]. Apfelbeck et al. [38] utilised image fusion by combining MRI information with real-time scanned ultrasound images to assess HIFU-ablated tissue in prostate cancer. Pathological evidence is necessary to ensure negative margins; thus, puncture pathology core-needle biopsy could be the less invasive option for follow-up [17]. Most studies followed the treat-and-resect protocol, except for Wu et al. [21], in which the pathological evaluation was performed using serial core biopsies at 2 weeks, 3 months, 6 months, and 12 months with the HIFU-treated tumour left in situ. In future studies where the HIFU-treated tumour is left in situ, the combination of US, contrast-enhanced MRI, and multiple puncture pathologies may be the best strategy for evaluating residual tumours [17].

Half of breast cancers are non-palpable, multi-centric breast cancer lesions. The use of HIFU ablation for this category of breast cancer lesion has not been studied extensively. Future clinical studies should include non-palpable, multi-focal breast lesions, adopting the treat-and-resect protocol where HIFU ablation is followed by surgery, allowing histopathological assessment of the resected breast tissue [17]. Additionally, different MRI methods should be studied for the larger variety of breast cancer types to fully assess their suitability as biomarkers for HIFU treatment evaluation [36].

## 5. Conclusions

HIFU ablation can induce tumour coagulation necrosis in localised breast tumours via one session, halting the growth without damaging the surrounding healthy breast tissue. There is potential for HIFU therapy to become an alternative therapy or to replace breast conservation surgery for a specific group of early breast cancer cases. The correct ablation margin is essential to establish, as it relates to long-term survival and local recurrence. Neither MRI nor diagnostic US have satisfactory sensitivity for precise visualisation of tumour margins. MRI may be the superior option given that several MRI contrast parameters demonstrated higher sensitivity for HIFU-ablated tissues. Multiparametric MRI evaluation shows potential for accurate and automatic segmentation of non-viable and viable tissue post-HIFU therapy. Overall, contrast-enhanced MRI is more precise and reproducible in determining lesion location, size, number, and borders of breast cancer. This area needs to be further explored in order to ensure tumour clearance with tumour-free margins while preserving healthy breast tissue and maintaining a cosmetically acceptable breast shape. Image fusion with ultrasound and MRI, alongside biopsies, may be the preferred option for evaluation of breast tumour clearance and follow-up after HIFU therapy.

Also, there is strict patient selection criteria for this novel therapy. The role of HIFU therapy in breast cancer management is currently specific to single palpable breast tumours, with very little evidence of its usage in different types of breast cancer tumours. There is limited applicability of HIFU therapy for breast cancer at present.

This present study suggests that HIFU therapy can become an excellent adjunctive therapy for multimodal breast cancer treatment, provided that there is more consistent evidence of its effectiveness in inducing complete coagulation necrosis. The question of whether this non-invasive therapy can replace breast-conserving surgery is currently inconclusive due to the lack of clinical evidence. Therefore, further experimental studies on the clinical application of HIFU therapy for breast cancer alongside its long-term outcomes need to be conducted. Also, a reliable method for assessing tumour margin and detecting residual tumour after HIFU ablation needs to be established. Once there is greater evidence of the clinical practice of HIFU therapy as part of conventional breast cancer management and technical advancement, standardised guidelines and treatment protocols can be produced. Overall, the current evidence has demonstrated that HIFU is a potential alternative therapy for treating breast cancer with the advantages of being a non-invasive, extracorporeal, non-irradiating modality that is feasible, safe, and repeatable. However, the key issues currently raise concerns about the effectiveness and reliability of HIFU therapy in achieving the desired outcome that is comparable to the existing breast conserving treatment. HIFU therapy remains in its clinical experimental stage due to the very low amount of high-quality randomised control trials.

## Figures and Tables

**Figure 1 diagnostics-13-02595-f001:**
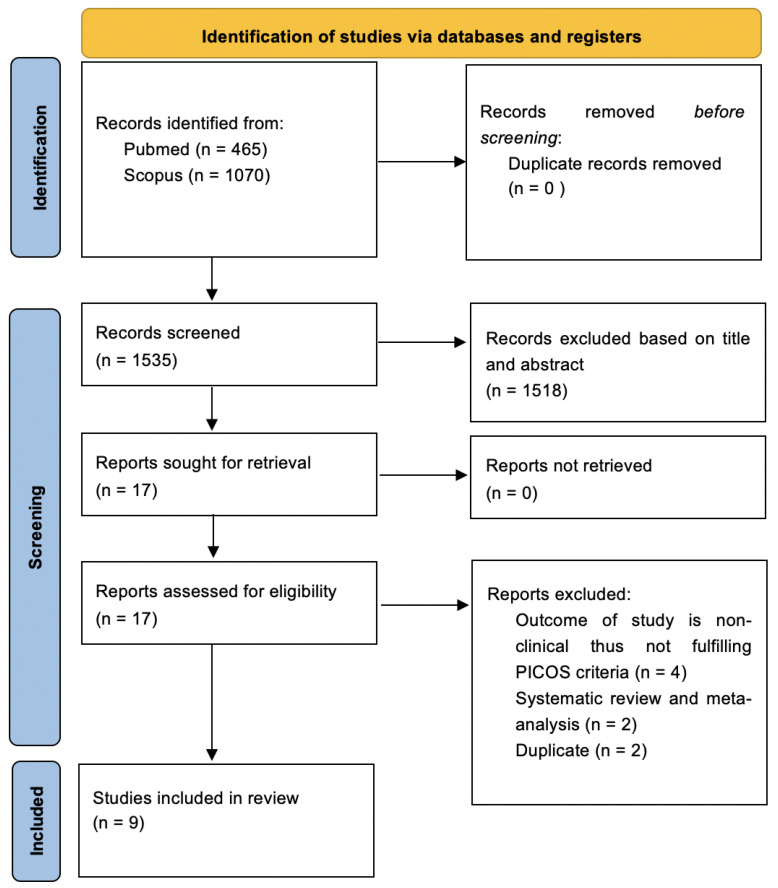
Search strategy based on PRISMA flow diagram for new systematic reviews, which included searches of databases.

**Table 1 diagnostics-13-02595-t001:** PICOS criteria for inclusion in the systematic review.

PICOS Criteria	
P—patient	Women aged 18 and above with primary breast cancer
I—intervention	High-intensity focused ultrasound (HIFU) therapy
C—comparison	In addition to conventional management (surgery +/− radiotherapy +/− chemotherapy +/− endocrine therapy)
O—outcome	Outcome (radiological, histopathological), survival rate, recurrence rate, HIFU complications, cosmesis
S—the type of study	Exclude studies with no statistical comparisons (case study or case series), consisting of less than ten patients, reviews, editorials, conference papers, more than 20 years old, non-English, and non-human studies

**Table 2 diagnostics-13-02595-t002:** Description of studies utilising high-intensity focused ultrasound (HIFU) to treat breast cancer.

Author(s), Year, and Country	Study Design	Demographic Information	Image Guidance Modality	Type of Breast Cancer
(Guan et al., 2016) [17] China	RCT HIFU + modified radical mastectomy (after 2 weeks) vs. Modified radical mastectomy alone	Total patients: 50 Control group (*n* = 25) Mean age: 45 (25–65) HIFU group (*n* = 25) Mean age: 48 (22–63)	US-guided HIFU	Invasive breast cancer (T1-2, N0-2, M0) Stage: I, IIa, and IIb
(Zippel et al., 2005) [18] Israel	Prospective Preliminary phase 1 trial	Total patients: 10 Mean age: 56 (45–72)	MRI-guided HIFU (MRgFUS)	Invasive breast cancer Stage: Unknown
(Furusawa et al., 2006) [19] Japan	Prospective	Total patients: 30 (28 patients evaluated) Mean age: 56.9 (41–79)	MRI-guided HIFU (MRgFUS)	Invasive breast cancer (T1–2, N0–2, M0) Stage: Unknown
(Gianfelice et al., 2003) [20] Canada	Prospective	Total patients: 12 Mean age: 60 (45–74)	MRI-guided HIFU (MRgFUS)	Invasive breast cancer Stage: Unknown
(Wu et al., 2005) [21] China	Prospective HIFU +/− axillary node dissection (*n* = 5) + adjuvant chemotherapy (*n* = 22) + adjuvant radiotherapy (*n* = 22) + adjuvant Tamoxifen (*n* = 22)	Total patients: 22 Mean age: 48.6 (36–68)	US-guided HIFU	Invasive, non-invasive, unidentified breast cancer Stage: I, IIa, IIb, and IV
(Wu et al., 2006) [22] China	Prospective HIFU + modified radical mastectomy (after 7–14 days)	Total patients: 23 Mean age: 46.5	US-guided HIFU	Stage: I and II
(Merckel et al., 2016) [23] Netherlands	Prospective Partial HIFU + surgical resection (2–10 days)	Total patients: 10 Mean age: 54.8	MRI-guided HIFU (MRgFUS)	Invasive breast cancer (T1-2)
(Wu et al., 2007) [24] China	Prospective HIFU + modified radical mastectomy (1–2 weeks) +/− chemotherapy/radiotherapy/hormonal therapy	Total patients: 23 Mean age: 46.5	US-guided HIFU	Invasive, non-invasive, breast cancer Stage: I and II
(Wu et, al., 2003) [25] China	RCT (clinical trial phase II) HIFU + modified radical mastectomy (after 1–2 weeks) vs. Modified radical mastectomy alone	Total patients: 48 Control group (*n* = 25) Mean age: 45.5 HIFU group (*n* = 23) Mean age: 46.5	US-guided HIFU	Invasive breast cancer (T1–2, N0–2, M0) Stage: I, II, and III

**Table 3 diagnostics-13-02595-t003:** Description of studies utilising high-intensity focused ultrasound (HIFU) to treat breast cancer.

Author(s), Year, and Country	Tumour Size (cm)	Surgical Resection	Clinical Evaluation/ HIFU Complication	Radiological Evaluation	Pathological Evaluation	Outcomes
(Guan et al., 2016) [17] China	HIFU: 2.1 to 4.8 Control: 2.3 to 4.5	14 days	oedema, pain/tenderness/discomfort (*n* = 11, 44%) mild fever over 38 within 48 h (*n* = 3, 12%)	-	HE staining UEAI staining Victoria blue and Ponceau’s histochemical staining	Complete necrosis = 100% Death = 5% (*n* = 1) Rate of local recurrence = 9% (*n* = 2) 5-year disease-free survival = 95%
(Zippel et al., 2005) [18] Israel	2.2 (mean)	7 to 10 days	2nd-degree burn (*n* = 1) pain	MRI	Unknown	Complete necrosis = 25%
(Furusawa et al., 2006) [19] Japan	0.5 to 2.5 1.3 (mean)	5 to 23 days	3rd-degree skin burns (*n* = 1) claustrophobia (*n* = 1) allergic reaction to plastic material of the patient’s interface (*n* = 1) sonication-related pain requiring NSAID (*n* = 2) shoulder pain (*n* = 1)	MRI	HE staining	Complete necrosis = 95% Mean necrosis = 96.9%
(Gianfelice et al., 2003) [20] Canada	0.11 to 8.80 cm^3^ (volume)	Unknown	slight pain (*n* = 4, 33%) moderate pain (*n* = 8, 67%) tenderness (*n* = 3, 25%) 2nd-degree skin burns (*n* = 2, 17%)	MRI	HE staining	Complete necrosis = 17% Mark 1 group (*n* = 3)—mean of 43.3% cancer tissue necrosed Mark 2 group (*n* = 9)—mean of 88.3% cancer tissue necrosed
(Wu et, al., 2005) [21] China	2.0 to 4.8 3.4 (mean)	Only axillary node dissection at 4 to 8 weeks (*n* = 5)	local mammary oedema five-point scale cosmetic evaluation—94% (good-excellent), 6% (acceptable)	B-mode US Colour Doppler US SPECT MRI	Core biopsy + HE staining	Complete necrosis = 100% 5 y Disease-free survival = 95% 5 y Recurrence-free survival = 89%
(Wu et, al., 2006) [22] China	2.0 to 4.7 3.1 (mean)	7 to 14 days	local mammary oedema (*n* = 23) mild local pain, warmth, and sensation of heaviness in the treated breast (*n* = 14) required analgesia (*n* = 4)	-	HE staining NADH-diaphorase	Complete necrosis = 91.3%
(Merckel et, al., 2016) [23] Netherlands	3 to 5	2 to 10 days	no pain (*n* = 8) pain score 4 and 5 (*n* = 2)	MRI	HE staining	Complete necrosis = 60%
(Wu et, al., 2007) [24] China	2.0 to 4.7 3.1 (mean)	7 to 14 days	local mammary oedema palpable lump larger than lesion size sensation of heaviness mild local pain (*n* = 14) required analgesia (*n* = 4) minimal skin burn (*n* = 1)	MRI	HE staining TCC staining TUNEL method	Complete necrosis = 100%
(Wu et, al., 2003) [25] China	HIFU: 1.8 to 5.6 3.5 (mean) Control: 2.0 to 4.7 3.1 (mean)	7 to 14 days	local mammary oedema sensation of heaviness warmth mild local pain required analgesia (*n* = 4) minimal skin burns (*n* = 1)	MRI	HE staining Immunohistochemical staining for PCNA, CD44v6, MMP-9	Complete necrosis = 100%

**Table 4 diagnostics-13-02595-t004:** Time distribution of MRI-guided HIFU by Meckel et al. [23].

Stage of Procedure	Time in Min, Mean ± SD (Range)
Positioning on treatment table (including MR imaging until contrast injection)	25 ± 10 (5–39)
Pre-treatment imaging from contrast injection to the first (test) sonication	59 ± 27 (32–106)
Treatment time (from first to last sonication)	46 ± 17 (12–75)
Post-treatment imaging after the last sonication	14 ± 3 (7–19)
Overall procedure time	145 ± 29 (96–210)
Overall sonication time	1.7 ± 0.8 (0.3–2.6)

## Data Availability

Not applicable.

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
