# Peer review of "The Applications of High-Intensity Focused Ultrasound (HIFU) Ablative Therapy in the Treatment of Primary Breast Cancer: A Systematic Review"

_diagnostics, 2023, doi:10.3390/diagnostics13152595_

Round 1
Reviewer 1 Report
The paper summarizes a study that evaluated the use of high-intensity focused ultrasound (HIFU) ablative therapy in the treatment of primary breast cancer. The researchers conducted a search of the PubMed and Scopus databases following PRISMA guidelines and identified nine studies that met their inclusion criteria. The included studies consisted of two randomized controlled trials and seven non-randomized clinical trials. Currently, HIFU therapy is limited to single, palpable breast tumors. The authors emphasize the need for more extensive and randomized clinical trials to evaluate the efficacy of HIFU therapy for breast cancer, especially in cases where the tumor is left in situ. While the study highlights the potential benefits and favorable safety profile of HIFU ablative therapy for primary breast cancer, there are several drawbacks and limitations that should be considered:
-
Inconsistent evidence of complete coagulation necrosis: The study notes that there was variable evidence of complete coagulation necrosis in the HIFU-treated tumors across the included studies. This inconsistency raises concerns about the effectiveness and reliability of HIFU therapy in achieving the desired treatment outcome.
-
Limited applicability: HIFU therapy is currently limited to single, palpable breast tumors. This restricts its use to a specific subset of patients and excludes those with multiple tumors or non-palpable lesions. The study suggests the need for more research to expand the applicability of HIFU therapy to a broader range of breast cancer cases.
-
Limited evidence base: Although the study included nine studies, only two of them were randomized controlled trials (RCTs). The majority of the included studies were non-randomized clinical trials. This limited evidence base and the lack of high-quality RCTs weaken the overall strength of the findings and make it difficult to draw definitive conclusions about the efficacy and safety of HIFU ablative therapy for breast cancer.
Author Response
Thank you for the constructive comments.
I have added a few points regarding the radiological evaluation of HIFU therapy for breast cancer and revised the conclusion of this study.
The amendment are highlighted in yellow.
Reviewer 2 Report
The authors are adressing an important subject, the role of HIFU in the local therapy of breast cancer. They have reviewed the most current literature and they are elaborating not only on the benefits of HIFU but also on the critical issues. They are adressing the very important question of the prediction of success by imaging that is one one the key problems in this field. However, I would recommend to put much more emphasis on this point. Furthermore they are speculating that HIFU could replace all local therapies in selected cases. There are literally no data supporting this speculation. As long as breast conserving therapy means breast conserving surgery plus whole breast irradiation HIFU can only replace the surgical component. And last but most important: The conclusion should be much clearer that HIFU at the moment has no place in the therapy of breast cancer and that without randomized clinical trials versus the SoC this will not be the case in the future. HIFU is nothing more but "promising".
Author Response

(The authors gave the same response as above.)

Round 2
Reviewer 1 Report
Accept in present form